# Modification of Fabrication Process for Prolonged Nitrogen Release of Lignin–Montmorillonite Biocomposite Encapsulated Urea

**DOI:** 10.3390/nano13121889

**Published:** 2023-06-19

**Authors:** Mohamed I. D. Helal, Zhaohui Tong, Hassan A. Khater, Muhammad A. Fathy, Fatma E. Ibrahim, Yuncong Li, Noha H. Abdelkader

**Affiliations:** 1Soil Sciences Department, Faculty of Agriculture, Cairo University, Giza 12613, Egypt; hakhater2000@agr.cu.edu.eg (H.A.K.); mohamed.fathy@agr.cu.edu.eg (M.A.F.); zahraa19_92@hotmail.com (F.E.I.); noha.abdulkader@agr.cu.edu.eg (N.H.A.); 2School of Chemical & Biomolecular Engineering, Georgia Institute of Technology, Atlanta, GA 30332, USA; zt7@gatech.edu; 3Department of Soil and Water Sciences, Tropical Research and Education Center, Institute of Food and Agricultural Science (IFAS), University of Florida, Homestead, FL 33031, USA; yunli@ufl.edu

**Keywords:** lignin–montmorillonite encapsulated urea, controlled release urea (CRU), high nitrogen loading, nitrogen release rate in water, prolonged nitrogen release period in soil

## Abstract

Crop production faces challenges in achieving high fertilizer use efficiency. To address this issue, slow-release fertilizers (SRFs) have emerged as effective solutions to minimize nutrient losses caused by leaching, runoff, and volatilization. In addition, replacing petroleum-based synthetic polymers with biopolymers for SRFs offers substantial benefits in terms of sustainability of crop production and soil quality preservation, as biopolymers are biodegradable and environmentally friendly. This study focuses on modifying a fabrication process to develop a bio-composite comprising biowaste lignin and low-cost montmorillonite clay mineral for encapsulating urea to create a controllable release fertilizer (CRU) with a prolonged nitrogen release function. CRUs containing high N contents of 20 to 30 wt.% were successfully and extensively characterized using X-Ray diffraction (XRD), Fourier-transform infrared spectroscopy (FTIR), and Scanning Electron Microscopy (SEM). The results showed that the releases of N from CRUs in water and soil extended to considerably long periods of 20 and 32 days, respectively. The significance of this research is the production of CRU beads that contain high N percentages and have a high soil residence period. These beads can enhance plant nitrogen utilization efficiency, reduce fertilizer consumption, and ultimately contribute to agricultural production.

## 1. Introduction

Improving fertilizer use efficiency has drawn significant attention due to the increasing demand for food products to satisfy global population growth. About 40–70% of nitrogen (N) applied as conventional fertilizers is lost through various pathways, such as leaching and volatilization [1,2,3]. Recently, controlled-release fertilizers have been intensively studied because of their ability to reduce nutrient loss by encapsulating urea in the coating materials. These coatings play a crucial role in slowing down the dissolution rate of urea in the soil. The rate of urea dissolution and release is influenced by factors such as the types and thickness of coating materials and temperature [4].

Slow-release fertilizers (SRFs) have been specifically designed to delay nutrient release in synchrony with the needs of plants for nutrients, consequently enhancing nutrient use efficiency and improving yields [5]. The early stage of slow-release fertilizers (SRFs) focuses on coating fertilizers with sulfur and/or polymers. However, sulfur coating, while relatively inexpensive, requires a thick layer of sulfur and wax (~15–30 wt.% of SRF), which provides minimal benefit to plants. Sulfur coatings are prone to cracking, have low biodegradability, lack flexibility, and often fail to completely cover the fertilizer. Additionally, sulfur-coated fertilizers exhibit rapid and uncontrollable nutrient release rates [6]. Thus, there is a pressing need to develop new coating techniques that can further slow the nutrient release, ultimately improving fertilizer use efficiencies. Due to the unique properties of nanomaterials, incorporating nanomaterials into the fertilizer coating design has great potential in improving nutrient use efficiency. The nanocomposite materials may increase the physical properties of the coating layer [7,8,9].

Sustainable agriculture requires the use of nontoxic, biodegradable, and biomass-derived materials, which reduce negative impacts on the environment (e.g., greenhouse gas emission and air pollution) and leave no waste in our ecosystem (soil and water) [10]. Lignin, the most abundant aromatic biopolymer in nature, possesses a heterogeneous structure and exhibits excellent anti-microbial activity [11,12]. Lignin offers advantages such as low toxicity, high biodegradability, stability, cost-effectiveness, and strong reactivity owing to its aromatic nature and functional groups. These characteristics position them as potential high-performance adsorbents, thereby serving as promising chemical delivery systems [13,14]. However, to make them practical for various applications, improvements in adsorption capacity and rate are required. Chemical modification, functionalization, and hybrid nanocomposites, including crosslinking, are commonly employed strategies to enhance their adsorption properties, cation exchange capacity, stability, and durability. Previous studies have utilized a mixture of lignin and rosins to coat the soluble fertilizer (e.g., urea) [15,16]. In the present study, montmorillonite clay mineral was selected as an inorganic base due to its low cost, being environmentally friendly, its layered structure, and its unique mechanical and barrier properties at a nanometer scale. This research aims to further slowdown the rate of nitrogen release from the synthesized CRU and increase the soil residence time and plant nitrogen use efficiency. Two significant modifications were introduced in the original method [17]. Firstly, urea solution was added sequentially to the modified montmorillonite suspension before the addition of sodium alginate. This sequential addition allowed for the diffusion and penetration of urea molecules within the interlayers of the modified montmorillonite. Secondly, the concentration of the modified montmorillonite suspension was increased from 2% to 5% to enhance the thickness and hardness of the coating, thereby prolonging soil residence time and regulating the nitrogen release rate. The prepared controlled-release urea (CRU) was characterized, and its performance in reducing nutrient release rates in water and soil was evaluated.

## 2. Materials and Methods

Materials: Black liquor (bio-refinery residues) was provided by Misr Edfo, a paper and printing company in Egypt. Four types of biomasses of agricultural residues (Alf Alfa, hard date palm fibrillum (HDPF), soft date palm fibrillum (SDPF), and sawdust (S. dust)) were collected from the experimental station of the Faculty of Agriculture and carpenter workshop, respectively. The following reagents—commercial lignin, montmorillonite clay mineral, sodium alginate, sodium hydroxide (NaOH), and di-hydrated calcium chloride (CaCl_2_·2H_2_O)—used in this research were purchased from Alpha Chemika (Mumbai, India). The montmorillonite clay mineral was used without purification. The commercial urea (NH_2_)_2_CO in the form of granules (46% N) used in this research was purchased from a local agro-market in Giza city, Egypt.

### 2.1. Preparation of Lignin

Alkaline Lignin was prepared using black liquor, which is a byproduct of the paper manufacturing process, and agricultural residues, by being treated with 1.5 M NaOH at a ratio of 1:10 (residue: NaOH solution) at 70 °C for 120 min according to the kraft method [18,19,20]. Organsolv lignin was extracted from four agricultural residues—Alf Alfa, hard date palm fibrillum (HDPF), soft date palm fibrillum (SDPF), and saw dust (S. dust)—by treating them with formic and acetic acids using the Organosolv solution [21].

### 2.2. Preparation of a Modified Lignin–Montmorillonite Bio-Composite-Based Encapsulated Urea (CRU)

The lignin–montmorillonite biocomposite CRU was prepared according to the methods described in previous research [17,22]. However, we made two main modifications to further slow the nitrogen release rate down. In the first one, a urea-saturated solution (pH 6.5) was initially added to the stable montmorillonite suspension before alginate and was stirred for 24 h to ensure the complete penetration of urea molecules interlayering the modified montmorillonite that led to the further slowdown of N release. The second one used a higher concentration of the modified montmorillonite suspension of 5%, instead of 2%, to increase the thickness and hardness of the coating, which lead to an increase in its soil residence time, ultimately reducing the nitrogen release rate.

#### 2.2.1. Synthesis of Epoxypropyl Trimethylammonium Chloride (ETAC) and Quaternary Ammonium Lignin (QAL)

In an ice–salt bath (NaCl/ice = 1:3 by weight), trimethylamine (TMA) and epichlorohydrin were mixed in a three-neck conical flask installed with a condenser, at a molar ratio of 10:7 and stirred for 1 h and were then left overnight for the complete reaction and formation of ETAC. A beaker containing 2.5 g of lignin dissolved into 25 mL of 20% *w*/*v* NaOH was placed in a water bath at 80 °C for 20 min. Following that, 26 mL of ETAC was added to the mixture and reacted for 5 h at constant stirring until a brown–red emulsion was obtained. The obtained product (QAL) was then dried under a vacuum and stored in a refrigerator at 4 °C.

#### 2.2.2. Preparation of QAL-Modified Montmorillonite Clay Mineral

A weight of montmorillonite clay mineral (5 g) was added to 400 mL of deionized water (DW) in a beaker. It was stirred until a homogeneous suspension was obtained. A suspension containing 10 g QAL and 100 mL DW was added to the montmorillonite clay mineral suspension and stirred overnight. The modified montmorillonite was separated from the aqueous phase by centrifugation at 3500 rpm. It was then washed with DW, then ethanol, and separated by centrifugation until no chloride was detected by using silver nitrate method. After that, the product was freeze-dried and stored in a refrigerator at 4 °C.

#### 2.2.3. Preparation of a Modified Lignin–Montmorillonite Bio-Composite-Based Encapsulated Urea (CRU)

Suspensions of 2 and 5% *w*/*v* of QAL-modified montmorillonite were prepared and stirred until stable. The second modification was that urea solution was added to the QAL-modified montmorillonite and stirred overnight to ensure the diffusion and interlayer penetration of the urea molecule into modified montmorillonite prior to adding sodium alginate. A saturated urea solution was prepared by dissolving 20 g of urea in 20 mL DW. It was heated to obtain a clear solution, then added to the suspension of QAL-modified montmorillonite and stirred overnight. After that, the mixture was heated to 80 °C and 2% *w*/*v* of sodium alginate solution was added and stirred to form a gel. The formed gel was kept in a refrigerator for 24 h for stabilization, then dropped into heated 4% calcium chloride solution to form the beads of CRU. The beads were separated, then dried in an oven at 55 °C, and used for characterization, measurements of N release rate, and the greenhouse experiment.

The five types of CRU differed in the coating type as shown in Figure 1; alginate only (product 5) or modified montmorillonite (MMT) at two suspension concentrations of 2 and 5% with the addition of alginate for the other four products. Products 1 and 2 had MMT of 2%, but in the former one alginate was added before urea. In the latter one, urea was added before alginate. Products 3 and 4 had MMT of 5%, but in the former one alginate was added before urea, whereas in the latter one urea was added before alginate.

### 2.3. The Characterization of the Montmorillonite Clay Mineral, Lignin, and QAL-Modified Montmorillonite

Clay, lignin and lignin–montmorillonite bio-composite encapsulated urea (CRU), were characterized using Fourier-transformer infrared spectroscopy “FTIR” (FTIR spectrometer, JASCO, FT/IR-4100 type A, 399.193–4000.6 cm^−1^) to explore the chemical bonds and to trace the changes in chemical groups as a result of binding between lignin and montmorillonite clay mineral as well as urea and alginate. X-Ray Diffraction (XRD, PAN analytical Empyrean, Almelo, The Netherlands) was used to explore mineral composition of montmorillonite used in the preparation of beads of CRU. Scanning Electron Microscopy (SEM) (Field Emission Scanning Electron Microscope, Model “Zeiss Sigma 500 VP Analytical FE-SEM Carl”, Zeiss Company, Oberkochen, Germany) was used to observe the CRU beads morphologies and coating porosity. The scale of the synthesized product was scanned between 5–30 μm.

### 2.4. Dissolution of Lignin–Montmorillonite Bio-Composite-Based Encapsulated Urea in Water and the Release of Nitrogen

CRU beads (0.2 g) were weighed and placed in 50 mL falcon tubes. Following that, 20 mL of DW was sequentially added to the beads and left for different equilibrium periods—1, 6, 24, 48, 96, 240, and 480 h. In all these successive extractions, the solutions were collected to determine the total N released from the CRU beads. All treatments were done in duplicates.

### 2.5. Solubility of Lignin–Montmorillonite Bio-Composite Encapsulated Urea and the Release of Nitrogen in the Soil

A mass of 0.5 g of CRU beads was placed in a glass column containing 40 g of clay loam soil which occupied a 30 cm depth of the column. The soil was collected from the eastern farm of the faculty of agriculture, Cairo University, air-dried, and sieved through 2 mm. The physical and chemical properties of the soil used in this study are pH (7.58), EC (2.8 dS m^−1^), organic carbon (0.96%), total carbonate content (2.7%), available macro-nutrients (111.4, 13.5, and 220 mg kg^−1^, for N, P, and K, respectively), coarse sand (4.4%), fine sand (31%), silt (27.3%), and clay (37.3%). The beads were mixed with the surface 2 cm layer of the soil in the column. Portions of water that were enough to saturate the soil (equal to the water-holding capacity of the soil sample) were added to each column. After complete penetration of water into the soil, 20 mL of DW was added to each column. The percolated water was collected and kept for the determination of total N. The first leachate represented the amount released in a period of 1 h. The soil was repeatedly treated with the same portion of water at 24, 96, 192, 288, 384, 480, 576, 672, and 768 h.

### 2.6. Analytical Procedures

#### 2.6.1. Determination of Total N in CRU Beads and in the Leachates of N Released in Water and Soil

Portions of 0.2 g (weight to the nearest four digits) were subjected to a wet digestion method using concentrated H_2_SO_4_ (98% *w*/*v*) and hydrogen peroxide (H_2_O_2_, 30%) [22]. Total N was determined in the acid digestion extract using the ammonia-distilling unit of Kjeldahl. The same method was used for the determination of total N released in water and soil leachates. The portions of water used for the dissolution of CRU beads and the leachates of the soil were collected. They were then subjected to wet digestion using the above-mentioned method.

#### 2.6.2. Determination of Total Organic Matter

Total organic matter (OM) content was determined in montmorillonite clay mineral, lignin-modified montmorillonite clay mineral, and the soil used in N release experiment using the Walkley and Black method [23]. This method is suitable for materials that contain a relatively low organic carbon content, e.g., montmorillonite clay mineral.

### 2.7. Determination of Total Carbonate Content (TCC)

Total carbonate content in montmorillonite clay mineral was determined using Collin’s calcimeter [24], which was then calculated as CaCO_3_.

## 3. Results and Discussion

### 3.1. Characterization of Montmorillonite Clay and CRU

The XRD analysis revealed that the montmorillonite clay mineral sample primarily consists of major triclinic and two types of monoclinic montmorillonite clay minerals with a high degree of crystallinity (Figure 1). The chemical analysis further indicated that the montmorillonite sample contains 0.2% organic matter (OM), and 2.1% total carbonate content (TCC). The presence of OM and TCC as cementing materials significantly influences the binding of montmorillonite layers, restricting their expansion. Additionally, soluble salts, particularly divalent calcium and magnesium salts, contribute to the coagulation of particles, thereby affecting the thorough penetration of quaternary ammonium lignin (QAL) into the interlayers of montmorillonite clay mineral. As the content of the QAL-penetrated interlayer montmorillonite clay increased, the durability of the lignin–montmorillonite biocomposite increased, resulting in an extension period of N release and enhanced nutrient use efficiency by the plant. Based on our findings, we suggest that removing salts and cementing materials from montmorillonite clay mineral using purification treatments [25] may enhance the complete expansion of the montmorillonite clay mineral layers, promoting full exfoliation and layer dispersion. This facilitates the penetration of QAL into interlayer spaces of the clay mineral, leading to the increasing strength and stability of the lignin–clay biocomposite.

Figure 2 presents the Fourier-transformer infrared spectroscopy (FTIR) spectrograph which shows several chemical bonds present on the surface of lignin. A broad band observed at 3400 cm^−1^ is attributed to the hydroxyl groups in phenolic and aliphatic structures [26]. The C-H stretching appeared at 2900 cm^−1^, indicating the present of aromatic methoxyl groups, methyl, and methylene groups [19]. In the carbonyl/carboxyl region, weak to medium bands were found near 1722 cm^−1^. The presence of carbonyl stretching in the organosolv lignin spectrum can be attributed to esterification between phenol and alcohol moieties of the propane chain, which occurs during the pulping process using formic acid [21,27]. The peak observed at 1500 cm^−1^ corresponds to C-C stretching in aromatic structures, representing the characteristic aromatic skeletal vibration of lignin’s aromatic rings [21,28]. The bands between 1020 and 1300 cm^−1^ indicate the presence of both syringyl and guaiacyl in the chemical structure of lignin [20,21]. It should be noted that, apart from the carbonyl group, the alkaline lignin (Edfo) exhibits the same chemical groups as the organosolv lignin.

The chemical compositions of lignin vary widely among plant families and species. However, all types of lignin consist of three monomers: p-coumaryl alcohol, coniferyl alcohol, and sinapyl alcohol. These monomers are from both C-O and C-C bonds, resulting in a heterogeneous structure and a three-dimensional structure [29]. The FTIR analysis showed that the surface chemical groups identified in all lignin types are mostly similar, with only minor differences observed between plant species sources and/or extraction methods. These chemical groups identified in our study align with the previous study [29], which reported that lignin had various functional groups such as phenolic hydroxyl, aliphatic hydroxyl, benzyl alcohol, noncyclic benzyl ether, carbonyl groups, and methoxyl groups. As no significant differences were observed among the six types of lignin tested, Edfo-lignin was used for preparing the CRU beads due to its affordability and ease of preparation. Additionally, it is an environmentally friendly option, as it is a waste product from the paper and printing company (Edfo, Egypt).

The FTIR spectrum in Figure 3a shows the existence of the surface chemical groups peaks of lignin, clay, and lignin–clay hybrid. The O-H, C-H sp3, C=C, C-C, and C-O appeared at 3400, 2900, 1680, 1467, and 1033 cm^−1^ for lignin, montmorillonite, and modified montmorillonite. The peak at 1200 could be attributed to C-N appearing in both lignin and modified montmorillonite. These conversions are attributed to the binding between lignin and montmorillonite to form lignin-modified montmorillonite. Comparing the strength and existence of the peaks of lignin with those of modified montmorillonite shows that the O-H and C-N groups of lignin were markedly reduced in modified montmorillonite and the C-H sp3, which only existed in lignin, appeared in modified montmorillonite. On the other hand, the C=C group disappeared in modified montmorillonite, and C-C and C=O were expended for modified montmorillonite being compared to lignin. For modified montmorillonite, the functional groups of O-H and C-O were reduced, the C=C disappeared, and the C-C had minimal change compared to that of montmorillonite. These modifications on surface chemical groups are good evidence for the binding between lignin and montmorillonite clay mineral, which is likely an ion exchange reaction between NH_4_*^+^* cation grafted on quaternary ammonium lignin, and hydrated exchangeable cations (e.g., Ca^2+^, Na^+^, Mg^2+^, … etc.) on the interlayer surface of montmorillonite to form a lignin–montmorillonite clay mineral bio-composite. A previous study investigated the modification of montmorillonite with lignin and showed that the presence of the C-H bond of lignin was observed in the modified montmorillonite, whereas it was absent in the unmodified montmorillonite [30].

Figure 3b showed the FTIR spectrograph of two types of controlled release urea (CRU). In the first type, urea was coated by modified montmorillonite and Na alginate (red line), whereas, in the second one, urea was coated by Na alginate only (black line). Five peaks are shown for the two types of coating at 3330, 3445, 1675, 1467, and 1152 cm^−1^, which are referred to as O-H, NH_2_, C=O, C-C, and C-N groups, respectively.

Scanning Electron Microscopy (SEM) images depict the surface morphology of lignin–montmorillonite bio-composite-based encapsulated urea (CRU beads) at various magnifications ranging from 2500 to 20,000× (Figure 4). The images reveal a striking similarity in the surface morphology of the samples. At the highest magnification, the core material appears to be smooth and exhibits a regular shape. On the other hand, the shell material, composed of modified montmorillonite, lignin, and alginate, displays an irregular and rough appearance with noticeable cracks and a few pores. These characteristics can be attributed to the production conditions of the bio-composite. The amplified images also indicate the presence of multiple layers, with the surface layer containing modified montmorillonite appearing rough and irregular. The cracks and pores observed in the underlayer are smaller compared to those on the surface layer. Similar observations were reported in our previous work on the surface of lignin–clay nanohybrid slow-release fertilizers [17]. The coating shell exhibited an irregular and rough texture with cracks, while the cross-section image revealed the well-matched aggregation of clay with alginate, resulting in a relatively uniform cross-section. Additionally, the combination of surface and cross-section images revealed the presence of thin white lines, a few nanometers wide, instead of the layered or particle structures typically observed in conventional slow-release fertilizers.

### 3.2. Total Nitrogen Contents in Lignin–Montmorillonite Bio-Composite-Based Encapsulated Urea (CRU)

All prepared CRU contained high percentages of N, as shown in Table 1, ranging from 20 to 30%, indicating their significance to be used as a high-quality nitrogenous fertilizer to feed the plants. Except for commercial urea (N ≈ 46%), and to a lesser degree ammonium nitrate (N ≈ 33%), the percentages of N in the prepared CRU were higher than all mineral nitrogenous fertilizers available in the agro-market. Fertilizers of high nutrient content are much preferable because they save on transportation and application costs [30]. Although substantial modifications were done to raise the hardness of the coating, hence the decrease N release rate and the increase CRU soil residence time, all used techniques had high efficiency in developing products containing high N percentages as shown in Figure 5.

The product with the highest N percentage was product No. 5, which had the thinnest coating that was formed solely from alginate. As the coating thickness decreased, the N content in the product increased, and vice versa. Conversely, the two products with the lowest N percentages (20.11% and 22.79%) were products No. 3 and 4, respectively, which had the thickest coating formed from 5% modified montmorillonite suspension with the addition of alginate. The decrease in the percentage of N in product 3 compared to product 4 can be attributed to the initial addition of alginate before urea during the preparation of CRU beads, which hinders the penetration of urea molecule into the interlayer montmorillonite clay mineral, resulting in reduced N content in the product. The initial addition of urea before alginate for the second one facilitates the penetration of urea molecule interlayer montmorillonite clay mineral, increasing the N content. On the other hand, the intermediate N percentages (24.66% and 26.84%) reported for products 1 and 2 could be attributed to the lower coating thickness that formed from 2% modified montmorillonite suspension and alginate, compared with that (5%) used for products 3 and 4. The reduction in N percentage in product 1 comparing to that of 2 was attributed to the initial addition of alginate before urea in product 1.

### 3.3. Dissolution and Release of N from Lignin–Montmorillonite Bio-Composite-Based Encapsulated Urea (CRU Beads)

#### 3.3.1. Solubility and Release of N in Water

The solubility and release of N from lignin–montmorillonite bio-composite controlled release encapsulated urea (CRU) in water were compared with commercial urea, as shown in Figure 6. The results showed a gradual release pattern of N from CRU over successive portions (time). In contrast, the commercial urea (blue line) rapidly dissolved in the first portion of water within 1 h of contact with particles. On the other hand, the release of N from CRU was significantly slower, extending over 20 days over seven water portions. Among the CRU products (1, 2, 3, 4, and 5), product 5 (red line) had the highest N release rate. The total content of N in product 5 was released approximately within 10 days (over six water portions), whereas products 1, 2, 3, and 4 released 76–90% of N during the same period. This difference can be attributed to the lower stability of the alginate coating used in product 5 compared to the lignin–montmorillonite bio-composite coating of the products 1, 2, 3, and 4.

The variation (76–90%) shown in N released from the four products (1, 2, 3, and 4), coated with modified lignin–montmorillonite bio-composite (after 10 days), could be partially attributed to the concentration of modified montmorillonite suspension, which was applied at two ratios (2% and 5%), and/or the pattern of sequential addition of alginate and urea during the preparation of CRU. Products 1 and 3 (azure and green lines, respectively), in which alginate was initially added before urea, showed a faster N release (90 and 85%, respectively) than those (78 and 76%, respectively) for products 2 and 4 (purple and orange lines, respectively), in which urea was initially added before alginate. After 20 days, the percentages of N released were 93% and 88% for the former, compared to 83% and 78% of the latter. Similarly, our previous study reported that the uncoated urea fertilizer was dissolved in water immediately and all the newly synthesized slow-release fertilizers (SRFs) with alginate, montmorillonite clay, lignin–clay nanohybrid, and PAA-coated lignin–clay nanohybrid had a slow-release capability [17]. The control (sodium alginate SRF) had a faster release rate (87%), while the PAA-coated lignin–clay nanohybrid SRF had the slowest release rate (74%) during the first five days [17].

#### 3.3.2. Solubility and Release of N in Soil

Figure 7 shows the pattern of N released from CRU in soil compared with commercial urea. The results showed gradual release patterns of N from CRU in soil, similar to its release in water. However, the release rates of N in soil were slower and extended over a period of 32 days compared to 20 days in water. In contrast, commercial urea was completely dissolved and released its N within 8 days. The release pattern of N from products 1, 2, 3, and 4 exhibited two distinct slopes. The first slope, lasting for 4 days, can be attributed to the release of N from the surface-bound urea molecules on lignin-modified montmorillonite. The second slope represents the controlled release of N that occurs over longer periods (32 days) and is associated with the diffusion of urea molecules penetrating the interlayer of lignin-modified bentonite.

Figure 7 also showed that product 5 (red line) with alginate coating exhibited the quickest N release rate among the four CRU products with lignin–montmorillonite bio-composite coating. After 12 days, the percentage of N released from product 5 was 89%, against 44–70% for the other 4 products. After 20 days, the total content of N in product 5 was completely released against 61–95% reported for the four products (1, 2, 3, and 4). These results showed considerably higher stability for the products that had lignin–montmorillonite bio-composite coating type than those that only had alginate coating.

Furthermore, the variations in N release (44–70%) observed among the four products with lignin–montmorillonite bio-composite coatings can be partially attributed to the concentration of modified montmorillonite suspension used (2% and 5%) and the sequential addition of alginate and urea during the preparation of CRU. Regarding products 1 and 3 (azure and green lines), in which alginate was initially added before urea, the percentages of N released after 12 days represented 70 and 62%, respectively, which were considerably higher than 49% and 44% of products 2 and 4 (purple and orange lines), in which urea was initially added before alginate. After 20 days, the percentages of N released were 95 and 89% for products 1 and 3, respectively, and 71% and 61%, for products 2 and 4, respectively. Although the total contents of N in products 1 and 3 were approximately released (100% and 95%) after 24 days, considerably lower percentages of N (82% and 72%, respectively) were released from the products 2 and 4. The release of N from products 2 and 4 continued for 32 days, for which the N released percentages were 99 and 98%, respectively. The obtained results proved the slow rate of N released from CRU that increases its residence time in soil. This caused an increase in the plant utilization efficiency of N. Our previous study reported that the uncoated urea dissolved within five days, which was slower than that in water [17]. The alginate-coated slow-release fertilizer (SRF) had a faster release rate (83%). In comparison, the PAA-coated lignin–clay nanohybrid/alginate beads had the slowest release rate (43%) during the first five days [17]. Additionally, a study of the release of N from single-layer bio-based polyurethane and multifunctional double ones of copper alginate and bio-polyurethane controlled release fertilizers reported that the release was sustained for 30 and 42 days, respectively [31].

## 4. Conclusions

In this research, we made two significant modifications to the method of synthesizing lignin–montmorillonite bio-composite encapsulated urea to further reduce the nitrogen release rate. The first modification involved the sequential addition of urea solution and sodium alginate, while the second modification utilized a higher concentration of modified montmorillonite suspension (5% instead of 2%). XRD analysis confirmed the crystallinity of the montmorillonite clay mineral, FTIR analysis revealed the surface chemical bonding modifications in the CRUs compared to lignin and modified montmorillonite, and SEM images depicted the morphologies of the CRUs, showing the presence of multiple layers. The prepared CRUs exhibited high nitrogen percentages ranging from 19.15 to 30.91 wt.%, making them strong candidates in the nitrogenous fertilizers market. Laboratory experiments demonstrated that the release of nitrogen from CRUs in water and soil extended over periods of 20 and 32 days, respectively. One important finding of this research was the successful synthesis of CRUs with significant nitrogen content and extended-release periods. These characteristics contribute to reduced nitrogen loss, enhanced plant nutrient utilization efficiency, and ultimately increased agricultural production. For future research, we recommend using purified samples of montmorillonite clay with low impurities such as organic matter, total calcium carbonate, and soluble salts. The use of pure montmorillonite samples allows for the complete expansion of montmorillonite layers, which enhances the interlayer penetration of quaternary ammonium lignin (QAL) and increases the durability of the modified lignin–montmorillonite bio-composite. This, in turn, leads to longer nitrogen release periods and improved plant nutrient utilization efficiency.

## Data Availability

The data presented in this study are available on request from the corresponding author.

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
