# Peer review of "Modification of Fabrication Process for Prolonged Nitrogen Release of Lignin–Montmorillonite Biocomposite Encapsulated Urea"

_nanomaterials, 2023, doi:10.3390/nano13121889_

Round 1
Reviewer 1 Report
Greetings, Editor thank you for providing me with the opportunity to review the article. I reviewed the article with ID = nanomaterials-2366594. Overall, the article structure and content are suitable for the Nanomaterials journal. I am pleased to send you major comments, there are SOME minor flaws. Please consider these comments/suggestions as listed below.
- The title seems very ok.
- Abstract also seems very good.
- Keywords are ok
- The introduction is perfectly managed to write. I really appreciate the efforts. But I am bit concern about nano-Lignin, its missing from introduction. Please add few lines about it.
- Several sentences in introduction are very long. Please rephrased it. The long sentences are not ideal to understand it. Please revise your paper accordingly since some issue occurs on several spots in the paper.
- Please check the abbreviations of words throughout the article. All should be consistent.
- The main objective of the work must be written on the more clear and more concise way at the end of introduction section.
- Please provide space between number and units. Please revise your paper accordingly since some issue occurs on several spots in the paper.
- Experimental section is fine.
- Section 2.1 Line 92, the existing reference 17 also need another reference. Cite this one, its exactly same to explain the process. (1) Preparation and characterization of nanosized lignin from oil palm (Elaeis guineensis) biomass as a novel emulsifying agent (2) Thermal degradation and kinetics stability studies of oil palm (Elaeis Guineensis) biomass-derived lignin nanoparticle and its application as an emulsifying agent.
- Figure 1, 2 , 3 and 4 quality is very poor. Provide original source images.
- So far the result and discussion is well explained but the comparison with literature is missing. Please add a critical discussion as a comparison with others.
- Please add a comparative profile section.
- Regarding the replications, authors confirmed that replications of experiment were carried out. However, these results are not shown in the manuscript, how many replicated were carried out by experiment? Results seem to be related to a unique experiment. Please, clarify whether the results of this document are from a single experiment or from an average resulting from replications. If replicated were carried out, the use of average data is required as well as the standard deviation in the results and figures shown throughout the manuscript. In case of showing only one replicate explain why only one is shown and include the standard deviations.
- Conclusion section is missing some perspective related to the future research work, quantify main research findings, and highlight relevance of the work with respect to the field aspect.
- To avoid grammar and linguistic mistakes, minor level English language should be thoroughly checked. Please revise your paper accordingly since several language issue occurs on several spots in the paper.
- Reference formatting need carefully revision. All must be consistent in one format. Please follow the journal guidelines.
Minor level
Reviewer 2 Report
The work is in an interesting area and the choice of urea as N source sound.
the paper requires extensive editing - poor wording often meant i did not understand and the wording lacked a scientific approach
there is no indication of how repeatable this work is -
although the lignin is mentioned in the abstract there is no indication of the use of alginate- a long timed coating agent
the nano - side of the study is not well characterized nothing about surface charge etc yes the particles have a nano size but in their use it is as an aggregate
the work should be done with more than one soil type - and even the clay loam used was not characterized
ureases could have influenced the N release assay -- no mention of controls for this

requires a strong edit sticky notes used to show places where the writing confused me
also many sticky notes on scientific issues
Reviewer 3 Report
In my opinion, you show really interesting results, but I encountered some difficulties reading your article. Mainly because I think a scheme showing the synthesis protocol of you material is missing:
for example, in your table 1, entry 5, you talk about alginate as a type of coating... but strictly alginate does not contain N atoms...and it's a coating agent of urea and not montmorillonite.. so you have to write U+Alg. So I understood better by reading your article in ACS sustainable Chem Eng. 2020. In addition, my other main remark concerns your discussion part which deserves to be improved: you repeat your conclusions a lot. Please be more concise especially for the part 3.3.1 and 3.3.2... check your paragraphs starting by "on the other hand"..
Then I have a few questions:
- part 2.2.3 you use 20 g of urea and 2%wt sodium alginate solution... but what is the ratio urea/alginate to get 30.00 wt% of N?
- Part 3.1 you show the crystal structure of montmorillonite then you talk about the chemical analysis and you refer to "our results also suggest" but what results? and related to what?could you check this point and rephrase your sentence please. Also, ref 24 is not about your results and you don't give any data proving that purifying your material improves its layer dispersion...
Your discussion regarding FTIR spectroscopy should be carefully reviewed:
- you need to simplify your discussion
-Fig 2 : You write that "regarding the other three groups C-O exists in the commercial lignin only, whereas the C-C exists in all types of lignin exept the commercial one". I understand that you are interpreting the spectra, but how could you explain that you have strictly no C-C bonds in your modified lignin montmorillonite if you also have a C-H bonds...so you have to rephrase that sentence...is not it ?
-In the same paragraph, you said that "no peak was noticed for the two groups NH2 at 3700 cm-1" talking about the modified monmorillonite, but these peaks are observed in the figure 3a ... so correct that sentence...
- Also, in my opinion, you should mention articles to corroborate your FTIR peaks assignments. (I found this article industrial crops & products 2019 111767 from Liu M. et al. " Synergistic effect of montmorillonite/lignin on improvement of water resistance and dimensional stability of Populus cathayana")
-It should be mentioned that the band at 516 cm-1 corresponds to the Al-O stretching and also the C-C 1510 cm-1 attributed to the aromatic groups of lignin.
-in the discussion of figure 3b, you mention a peak at 3645 cm-1. I think this is not the right value because it is not seen in this figure.
- Then you attributed the higher intensity of your modified montmorillonite + sodium alginate to the Na alginate layer only, because the bonds are different ...but the intensities of the IR peaks depend on the polarity of the bonds, so could you better explain this part?
Regarding the TEM discussion, lignin corresponds to a network so if you give a size it must be specified that it concerns only a few isolated spherical particles even if they do not correspond to the shape of the majority of the objects observed in the image. For me, it is better to mention that it is a network or a non-uniform objects...
Concerning the SEM images:
- I disagree with what you said about the surface of your material being smooth and having a regular shape...correct that...or provide details as to why you say that..
- Then you mention the authors of the article ref 20 by saying "they" but it's your article!!! so why didn't you say "in our previous work..." and the material is completely different to me, so this comparison of SEM is useless.
- You should put the SEM images of Montmorillonite in order to compare the surface modification when you run the modification with lignin
Concerning part 3.3.1, why do you observe a change in slope for the release of N after 4 days? could you explain this point.
This is why when I saw your figure 7 I think that these results are interesting because we see that the release of N is controlled because the slope is constant over time.
Sorry for all these comments, but in my opinion the discussion of this article should be reviewed carrefully.
Some details :
-in your experimetal part write mass or volume with number
-Part 2.2.1 which quantity of ETAC you use?
- part 2.2.2 homogeneous
- After Fig. 1 FTIR Fourier Transform
and spcetra or spectrum but not spectrograph
- in differents parts in the TEM and SEM interpretations you said "nano" ... should be nanometer
- part 3.2 the % are wt% because could be also mol%
-
Reviewer 4 Report
Review comments on manuscript (nanomaterials-2366594) entitled: Modification of Nano-lignin-montmorillonite Biocomposite Encapsulated Urea Fabrication Process for Prolonged Nitrogen Release. Manuscript aimed to observe the effects of NPs on nitrogen utilization efficiency. Paper is well written and could be interesting for nanomaterials readership. The authors have well designed experiment and encapsulated nitrogenous fertilizer with nano-biocomposite, however, the realistic application especially prolonged N release is not well supported with results, thus, the extensive experiment is required in soil as well their impacts soil microbiome with plants. After encapsulation, how CRUs behave on plant growth with compared to other fertilizers? The presentation of results has flaws, clear presentation are required with highlighting novelties as large number of studies are published on this aspect.
NA
Round 2
Reviewer 3 Report
You reply correctly to my previous comments.
Only 2 things can be improved :
- read your article carrefully because there are still quite a few typographical errors (for example word pasted in the summary "forencapsulating" introduction "desingnto" or errors like figure 1 "the synthesize protocol"...part 3.1 " degreee".. etc....
-apply the same color order for the caption of the figure 6 than for the figure 7 because your comments concerning the figure 6 are wrong in the text. Figure 7 is the correct color order to fit your comments.
Reviewer 4 Report
Current version is changed significantly and presentation of content and clearity of results is improved.
NA
